# Mechanistic Approaches to the Application of Nano-Zinc in the Poultry and Biomedical Industries: A Comprehensive Review of Future Perspectives and Challenges

**DOI:** 10.3390/molecules28031064

**Published:** 2023-01-20

**Authors:** Zohaib Younas, Zia Ur Rehman Mashwani, Ilyas Ahmad, Maarij Khan, Shah Zaman, Laraib Sawati

**Affiliations:** 1Department of Botany, Pir Mehr Ali Shah (PMAS)-Arid Agriculture University, Rawalpindi 46300, Pakistan; 2Department of Botany, University of Malakand, Chakdara 18800, Pakistan; 3Department of Chemical and Life Sciences, Qurtuba University of Science and Information Technology, Peshawar 25124, Pakistan; 4Institute of Biology/Plant Physiology, Humboldt-University Zü Berlin, 10115 Berlin, Germany

**Keywords:** bio-fortification, nano zinc, food application, antioxidant status, antimicrobial, drug delivery, bio-medical applications

## Abstract

Bio-fortification is a new, viable, cost-effective, and long-term method of administering crucial minerals to a populace with limited exposure to diversified foods and other nutritional regimens. Nanotechnology entities aid in the improvement of traditional nutraceutical absorption, digestibility, and bio-availability. Nano-applications are employed in poultry systems utilizing readily accessible instruments and processes that have no negative impact on animal health and welfare. Nanotechnology is a sophisticated innovation in the realm of biomedical engineering that is used to diagnose and cure various poultry ailments. In the 21st century, zinc nanoparticles had received a lot of considerable interest due to their unusual features. ZnO NPs exhibit antibacterial properties; however, the qualities of nanoparticles (NPs) vary with their size and structure, rendering them adaptable to diverse uses. ZnO NPs have shown remarkable promise in bio-imaging and drug delivery due to their high bio-compatibility. The green synthesized nanoparticles have robust biological activities and are used in a variety of biological applications across industries. The current review also discusses the formulation and recent advancements of zinc oxide nanoparticles from plant sources (such as leaves, stems, bark, roots, rhizomes, fruits, flowers, and seeds) and their anti-cancerous activities, activities in wound healing, and drug delivery, followed by a detailed discussion of their mechanisms of action.

## 1. Introduction

Increasing concerns have been raised about the alarming rise in chronic malnutrition. Aside from that, nearly two billion individuals endure “hidden hunger”, which is characterized by an insufficient consumption of basic minerals in their everyday diet, despite rising agricultural crop output [1]. Until now, our farming sector has not been structured to improve people’s health; rather, it has been focused solely on boosting grain output and crop yields. That policy has led to a dramatic increase in mineral deficit in cereal crops, resulting in increased micro-nutrient deficiencies in users. Agriculture has been shifting from producing more food crops to generating nutritionally rich food crops in adequate numbers [2].

Many recent technologies are being widely used in poultry production in terms of quantity and quality, such as nanoparticles [3,4]. The nanoparticles of ZnO rank third based upon size, possess multifunctional and physical properties, and can be synthesized easily. ZnONPs affect the growth performance and physiological state of livestock and poultry in a dose-dependent manner [5]. Nanoparticles may have the potential to efficiently deliver trace minerals to animals, increasing mineral bio-availability. Due to the small size of these particles as well as their high surface area to volume ratio, they have been shown to improve availability [6]. Besides offering large surfaces, nanoparticles have high catalytic activity and strong adsorption capacities [7]. Recent advances in poultry production have extensively explored the use of nanoparticles [3,4]. The synthesis of nanoparticles can be accomplished chemically, physically, or biologically. In contrast to physical and chemical methods that consume energy or require toxic solvents, biological synthesis is safe, clean, biocompatible, and eco-friendly, enabling metal ions to be reduced rapidly at room temperature [8]. 

Zinc is a trace element that is required by all living things (King, 2011). Animal-sourced feeds and many plants contain zinc, but they also typically contain substantial amounts of phytate. Six-zinc ions bind very tightly to phytates, preventing their absorption in the digestive tract [9]. Zinc can be found in poultry feeds, either in the form of inorganic feed grade zinc, zinc chloride, zinc oxide, or in chelated forms of organic acids and amino acids [10]. In chickens, zinc deficiency is known to cause a slow growth rate, shortened and thickened legs with enlarged hocks, and frizzed feathers (Nielsen, 2012). Living organisms are affected by this compound as it influences the immune system, nucleic acid synthesis, cell proliferation, protein synthesis, protein and carbohydrate metabolism, and enzymatic activity [11]. In the liver, zinc is a cofactor for those enzymes that produce alanine aminotransferase (ALT), gamma-glutamyl transferase (GGT), and aspartate aminotransferase (AST) [12]. 

NPs offer excellent properties such as large surface area, increased catalytic activity, and powerful adsorption capacity [12]. Feed supplement and additives have been effectively used in poultry ration for productivity and improved general health and well-being [13]. Bone growth and mineralization can be promoted by ZnONPs [14]. More than 300 enzymes use it for catalysis, structural roles, and regulatory functions such as the metabolism of substances, including carbohydrates, energy, protein, and nucleic acids [11,15]. Zn shortage also has an effect on epithelial quality because it plays a key part in protein synthesis (Tabatabaie et al., 2007). A variety of feed additives and supplements are used in poultry rations to improve productivity and health [13]. 

In recent years, zinc oxide nanoparticles (ZnONPs) have been receiving widespread attention in biological research because of their low toxicity, bio-compatibility, and chemical stability [16]. In addition to fulfilling all the needs of the body, these nanoparticles are environmentally friendly. As a feed supplement, Zn NPs increase body weight gain, feed conversion ratio, meat quality, and egg quantity [17]. The mineral zinc is important for bone development and for maintaining normal growth. It also helps restore damaged tissue [13]. The increase in zinc causes increases in feed production costs and zinc excretion in feces that contributes to pollution in the environment. It contributes to the imbalance of other elements and reduces vitamins [18]. Decreasing levels of zinc in the plasma has resulted in impaired physiological processes and a variety of liver diseases including cirrhosis and hepatitis [19].

## 2. Different Techniques for Nanoparticles Synthesis

ZnO is an inorganic compound with distinctive dimensions such as a semiconductor, a broad spectrum of radiation absorption, piezoelectricity, pyro-electricity, and high catalytic activity [20]. Moreover, owing to its non-toxic qualities, ZnO has been designated as “Generally Recognized as Safe” (GRAS) by the US Food and Drug Administration [21,22]. As a result, it is safe to use on humans and animals. Metal nanoparticles have gained popularity in recent years. Generally, this is because of their small particle size, which makes them more reactive. ZnONPs are conventionally synthesized via physical and chemical methods, which provide higher yields and better control over particle size. Nevertheless, such technologies are regarded as undesirable because of high capital costs, high energy needs, and the use of poisonous and harmful chemicals [22,23].

The above characteristics trigger serious tainting in the ecosystem. Furthermore, earlier research has shown that the chemical production of nanoparticles is unsafe and less biocompatible [24]. Additionally, their clinical and biological uses have been circumscribed. The NPs formed were physio-chemically analyzed to identify their features such as size, shape, surface charge, functional group, and purity [25]. As an outcome, there is a demand to canvass and appoint cleaner, ecologically safe, economically feasible, and biocompatible NP synthesis methods. The green process of NPs using biologically mediated approaches has gained popularity in recent years as a viable alternative to standard physical and chemical methods [26,27] Figure 1.

A variety of unicellular and multicellular organisms participate in the biological synthesis of metal and metal oxide nanoparticles, including bacteria, yeast, fungi, viruses, and algae [26,27]. These formulations are low-priced, non-toxic, and environmentally satisfactory. Microbes work as a small nano-factories for reducing metal ions into metal NPs, utilizing enzymes and other biomolecules they secrete or produce. Even so, only a few microbes are known to be capable of synthesizing ZnONPs. Hence, there is a need to explore more potential microbes for the synthesis of ZnONPs. Many different physical and chemical techniques have been used to produce nanoparticles, including vapor condensation, inter-ferometric lithography, physical fragmentation, sol-gel processing, solvent evaporation, and precipitation from micro-emulsions [28,29]. The hazardous chemicals that are usually utilized in chemical procedures includes triethyl amine, oleic acid, thioglycerol, and polyvinyl alcohol (PVA), which is primarily used as a capping and binding agent to regulate the size of nanoparticles and avoid clumping [30,31,32,33].

Several of these dangerous compounds may persist or be linked in the end result of the nanoparticles. Thus, the biological method for NP synthesis has sparked appreciable attention in the creation of metal and metal oxide NPs due to the use of less harmful chemicals that are both eco-friendly and energy efficient. ZnONPs are biologically manufactured utilizing physiologically active compounds from plants and microorganisms such as microbes, fungus, and yeast [34]. Nanotechnology is emerging because of its efficacy, eco-friendly procedures, low cost, simplicity, and mass productivity [35]. Compounds isolated from various sections of the plant, such as leaves, roots, stems, fruit, and flowers, are used in biological synthesis employing plant extracts. Some plant extracts include complex phytochemical substances, such as phenol, alcohol, terpenes, saponins, and protein, which function as reducing and stabilizing agents throughout the manufacturing process. As microbes can easily reproduce, they are a better choice than plants for biological synthesis [36]. In spite of these limitations, there are several downsides to isolating and screening prospective microorganisms. The primary disadvantage is that the synthesis procedures are not cost-effective since they are time-consuming and need the use of chemicals as growing media. The presence of microbe-derived enzymes, proteins, and other biomolecules is critical in the degradation of nanoparticles. 

Furthermore, the protein released by microorganisms may function as a capping agent, providing stability to the production of nanoparticles [37]. Generally, not all microorganisms can produce nanoparticles since each has a unique metabolic pathway and enzyme activity. Thus, in this regard, the selection of suitable microorganisms (independent of their enzyme activity and metabolic pathway) is critical for the formation of nanoparticles. Cultures are generally allowed to develop in the culture media. Furthermore, metal precursors are required for the biological production of metal and metal oxide nanoparticles, which are typically given in the form of soluble salts and precipitated in a solution containing microbial cells or biological component extracts from microorganisms [34].

The synthesis reaction is usually completed within minutes or hours depending on the culture conditions, resulting in a white deposit on the bottom of the flask or a change in the color of the suspension. As a result, this suggests that the change was effective [34]. Moreover, many factors such as temperature, pH, metal precursor concentration, and reaction time are significant in determining the rate of formation, yield, and morphology of NPs [38,39,40]. Nanoparticles exhibit immense properties such as antibacterial, antifungal, anticancer and antioxidant, but these properties are mainly affected by the size, morphology and conditions of their synthesis. Macromolecules, ions, pH, and temperature are the major factors which determine the physical properties of these nanoparticles [41]. For example, temperature is one of the most influencing factors in determining the physical characteristics of nanoparticles; a study conducted by Kredy et al. indicates that when AgNPs were synthesized at three different temperatures, i.e., 25 °C, 35 °C, and 45 °C, they showed different absorption peaks when observed in the UV–vis spectra. It was observed that the wavelength was higher at lower temperatures and shifted towards lower wavelengths at higher temperatures which confirms that high temperature results in comparatively smaller sized nanoparticles. This smaller size may be due to the fact that at higher temperature, the reactants are consumed rapidly which results in small sized nanoparticles [42]. A similar pattern was also observed under different pH conditions, when nanoparticles were synthesized at three different pH conditions, i.e., 4, 7, and 9. Results revealed that highest color change and intensity was observed at pH 9 while there was almost no activity reported at pH 3.0, which confirms that pH has also a definite effect on the synthesis and physical properties of nanoparticles [43]. Green synthesis is regarded as one of the most efficient and low-cost methods of producing fabricated NPs. These NPs are of various natures, with various phytochemical groups that add unique properties to the application of nanoparticles Figure 2. Other types of synthesis, such as physical and chemical synthesis, are hazardous to plants and animals [42].

## 3. Application of ZnO NPs

There are no free forms of Zn in nature, but it may be found in the form of other components such as zinc carbonate, zinc oxide, zinc sulfide, and zinc chloride. Zn is largely acquired via mining and metal smelters [44]. The electrolytic method can even make it in its axenic form. Zn ions are powerfully absorbable to the soil, are taken up by plant roots, ensuant in a range of 15 to 100 mg/kg concentration in plants. Zn compounds are employed in catalysts, fertilizer, charging cells, photographic paper, fabrics, medicinal and domestic uses, beauty products, paints, ceramics, and nutraceuticals [44,45]. 

Zn is a nutritionally captious element for human beings, farm animals, and plants since it is entangled in a variety of biological processes in all life forms. It is mandatory for proper growth and development, osteogenesis, protein synthesis, transcription regulation, cell division, and immune system function. It is also essential for birds, thus aiding suitable growth, feathering, hormone production, bone strength, protein synthesis, DNA synthesis, reproductive performance, and the optimal metabolism of enzymes [46].

## 4. Antioxidant Activity of Zn NPs and Their Mechanism

Several stresses have a damaging influence on broiler health and productivity including environmental, dietary, microbial, and managerial factors. All of these pressures are followed by oxidative stress. Oxidative stress in cells/tissues occurs as a result of an imbalance between free radical generation and natural antioxidant defense, and it causes lipid peroxidation, protein nitration, DNA damage, and apoptosis [47,48]. In living organisms, reactive free radicals such as ROS (reactive oxygen species) regularly generate during cellular metabolism. RNS and ROS are both signaling molecules that regulate homeostasis at certain levels. The excessive generation of ROS and RNS, or deficient scavenging, stimulates free radicals. ROS, which include superoxide, hydrogen peroxide, and hydroxyl radical radicals, are produced by oxygen metabolism and are often controlled by the rate of oxidant creation and rate of oxidant removal [49]. High temperature is a principal cause of ROS generation because it generates a redox imbalance boosting pro-oxidants over anti-oxidants. Symptoms of heat stress include nutrient deficiency, diminished growth efficiency, decreased immune function, hypoxia, and a greater likelihood of death [50,51]. 

Different proteins and enzymes are engaged to stop the overproduction of free radicals produced in the body of living organism during metabolism. A protein, such as metallothionein, is involved in boosting the immune system of an organism by scavenging OH^−^ hyrdroxyl ions [52]. Similar to MT (metallothionein), hypothalamic pituitary adrenal (HPA) releases corticosteroid from the adrenal gland that seems best at medicating the damage caused by environmental stresses [53,54]. By activating mitochondrial superoxide dismutase (MnSOD), a superoxide anion is transformed into hydrogen peroxide. CAT and GPX are two enzymes capable of detoxifying H_2_O_2_. As well as mono-amino oxidase, ketoglutarate dehydrogenase, glycerol phosphate dehydrogenase, and p66shc, other mitochondrial components are responsible for ROS formation [55]. In addition to p66Shc, another two proteins, p46Shc and p52 Shc, are also members of the ShcA protein family. p66Shc is an adaptor protein involved in apoptosis in mammals. It has a molecular mass of 66 kDa. Mitochondrial ROS can be generated by this protein [56].

Increased temperature significantly reduces the quality of chicken meat [57]. Poultry subjected to cyclic heat exposure had decreased crypt depth, mucosal area, intestinal epithelial cell damage, and villus height of the gastrointestinal tract, which has a deleterious influence on food absorption [58]. Numerous studies relate oxidative stress to heat stress, implying a simultaneous increase in cell death and ROS formation in certain cells [59]. Although small concentrations of reactive oxygen species is required for cellular signaling [60], an uncontrolled increase in ROS concentration results in a chain reaction which indiscreetly targets all the biological molecules of the cell I, e Lipid [61], proteins and also the control house of the cell, the DNA, which ultimately result in intrinsically triggered apoptosis. Heat stress results in elevated levels of superoxide anion (O^2−^), which occurs at specific sites of ETC in the cell. The generation of O^2−^ has also been observed in mitochondrial semiquinones, and it was observed that semiquinones radicals were observed in a hyperthermic rat cell [62,63]. Moreover, it was also observed that heat stress results in a smaller concentration of SOD-mRNA, cytoplasmic SOD protein and enzyme activity, resulting in a greater concentration of ROS in cells [64]. Furthermore, it has also been observed that heat stress results in the production of transitional metal ions, which can easily donate electrons to oxygen, resulting in increased levels of superoxide anions. Therefore, it can be concluded that heat stress works in both ways, on one side it enhances the production of ROS, and on the other side, it decreases the efficiency of the cellular antioxidant defense system, leading to higher ROS levels which ultimately affect the cellular machinery, leading to cell death [64].

There are many contaminants in poultry feeds/feed ingredients, including environmental toxins, bacterial and fungal toxins, and other contaminants that have negative impact on the gut health of poultry. Poultry feeds/feed components are frequently polluted with a wide range of environmental toxins, bacterial and fungal toxins, and other contaminants that are known to impact gut health. Stress-induced changes in cellular processes as well as intestinal barrier function are evidence of oxidative stress. Mycotoxins are strain-specific compounds generated by a broad variety of fungi, notably molds. Aflatoxin, zearalenone, deoxynivalenol, trichothecenes, fumonisin, T-2 toxin, and ochratoxin are examples of prevalent mycotoxins [64].

Table 1 shows a literature review regarding the antioxidant effect of zinc nanoparticles in different in vitro and in vivo studies. The production of oxidative stress in the GIT occurs when these toxins come into contact with the epithelial cells or during absorption. Prolonged exposure to even lower concentrations of mycotoxins has been demonstrated to have an influence on immune function and intestinal function, as well as impair blood phagocytic function in chicks [65]. 

Protracted arsenic (As) vulnerability causes lipid peroxidation, lowers antioxidants, and ultimately causes cell death in versatile bodily tissues of chicken [74]. An As and Cu mixture gives rise to inflammatory consequences in the intestinal mucosa [75]. In poorly ventilated poultry houses, ammonia (NH_3_) is a leading source of air contamination. High levels of NH_3_ results in an excess of creatine kinase enzymes that lessen the developmental rate, weight loss, villus height, crypt depth, and meliorate nutrient efficiency [76,77]. Longer ammonia consumption produces a variety of disorders and impedes the wellness of broiler chicken [78].

The GIT microbiota is mostly composed of microorganisms, fungus, and protozoa. The microbiota community fluctuates over the segment, having the greatest concentration towards the distal parts of the GIT [79]. In response to probiotics, intestinal epithelial cells produce ROS, which act as secondary messengers assisting cellular communication. Tight junctions separate intestinal epithelial cells from the barrier, preventing microbial infiltration [80]. Coccidiosis is one of the most frequent parasite infections that affect chickens. ROS are scavenged from the intracellular environment by superoxide dismutase, catalase, and glutathione peroxidase (GPX) [81]. Superoxide dismutase (SOD_1_ and SOD_2_) catalyzes the conversion of the superoxide anion (O_2_) to H_2_O_2_, which is then degraded into H_2_O and O_2_ by catalase, whereas GPX lowers lipid hydroperoxides by integrating glutathione [82].

Respective cells of the intestinal mucosa and sub-mucosal regions ship RNS derived from nitric oxide synthases (NOS). As arginine is metabolized to citrulline, the nitric oxide radical (NO•) is formed, which is pivotal to neurons and immune systems. The net outcome of lipid peroxidation produces 4-hydroxynonenal, which encourages oxidative damage to the cell membrane and impairs cell incitement and mitochondrial malfunction. The intestinal mucosa has a large antioxidant defense system, comprising enzymes (CAT, SOD, or GPX) and non-enzymatic scavengers such as glutathione, transitory ions (e.g., Fe^2+^, Cu^2+^), or flavonoids [83]. Glutathione and SOD are intracellular antioxidants which are primarily found in the gastrointestinal tract, and their quantity increases throughout intestinal growth [84]. 

Antioxidant compounds in the diet help lessen free radicals in the intestine and protect it from damage. Various studies reported that oxidative stress is prejudicial to the welfare of birds. As a result, developing a cost-effective technique to minimize oxidative stress is critical. Supplementation with vitamin C and E assists antioxidant potency and immunological regulation [85]. Alpha lipoic acid, a powerful antioxidant that is both fat-soluble and water-soluble, is known to protect the fowl intestine from severe oxidative damage. Equol, generated from the iso-flavonoid daidzein, a key isoflavone of soybean, has the ability to suppress oxidative adjustment caused by ROS [86]. Equol has the capability to switch on the expression of antioxidant genes by boosting the antioxidant enzyme level [87]. In order to reduce oxidative stress in the GIT, individual ingredients or combinations of ingredients can be used. Both avian and mammalian species require zinc (Zn) to grow, develop, reproduce, and carry out metabolic functions [88] Figure 3.

Zinc also serves structural and catalytic roles in metalloenzymes. Zn is a cofactor of Cu/Zn SOD, a significant antioxidant enzyme, and it also plays significant role in oxidative stress. Similarly, adding Zn oxide nanoparticles boosted broiler antioxidant potential, as evidenced by increased Cu/Zn SOD activity and decreased malondialdehyde (MDA) buildup in both the blood and liver [89]. Zinc is a trace element that is required by all living things [90]. As a result, zinc is required for regular growth and maintenance, repairing damaged tissues, and is essential for bone and feather formation [91].

## 5. Antibacterial Application of Zinc Nanoparticles

Earlier, people not familiar with the importance of nanotechnology used different byproducts (such as biomedicine) obtained from medicinal plants to treat various microbial diseases [92]. In late 20th century, antibiotics were used as therapeutic agents to treat bacterial infection caused by both types of bacterial strain, i.e., Gram-positive strain and Gram-negative strain, in poultry and other live stocks [93]. The use of antibiotics during feed ingestion improves the growth status and several physiological parameters in poultry. However, the regular intake of antibiotic not only lowers their effectiveness in the broiler but their consumption has become an alarming sign for humans. Bacteria are generally prokaryotes having limited organelle features rich in plasma membrane, cell wall and cytoplasm, whereas bacterial cytoplasm is rich in 70s ribosomal subunits and super-coiled helical genetic material [94]. 

The cell wall is the outer covering of both bacterial strains that is protective in function and also provides shape to bacterial cell. Gram-positive bacteria cell walls have a thick peptidoglycan layer and are composed of amino acids and polysaccharides. The Gram-negative bacteria cell wall is thin compared to Gram-positive bacteria and more devastating because of the extra protection of the lipopolysaccharides layer [95,96]. The cytoplasm of bacteria consists of polysaccharides, proteins, salts, minerals, and water. The super-coiled helical DNA monitors growth, protein synthesis, and enzymes metabolism [97]. Antibiotic suppress the process of translational regulation in bacteria by disrupting the assemblage of 50S and 30S ribosomal subunits. Thus, this activity inhibits the formation of specific proteins required for cellular metabolism [98].

Broiler chickens being exposed to high temperature results in ROS formation at the cellular level which interrupts enzymatic activities and can cause severe disease [51]. Other sources of free radical are produced in various ionic forms by the ingestion of toxicant or chemical compounds in poultry feed additives. Several studies illustrated that ROS, i.e., hydrogen peroxide, accelerated during exposure to continuous light exposure and even in dark conditions [99,100,101]. The use of antibacterial agents is an effective way to treat pathogen infections and restrict bacterial growth at optimum doses and have no side effects on the host body [102]. Zinc is an important mineral in the body of living organisms [103]. Evidence indicated that ZnONPs have great potential as antibacterial agents. ZnO nanoparticles inhibit the growth of bacterial strains such as E. coli and aureus. ZnONPs interact with the cell walls of microbes, discharging Zn^2+^ ions through channels, which, hence, results in the demolishing of bacterial cell viability [100,104,105,106,107].

Meanwhile, Padmavathy et al. proposed an association between photon reaction and antibacterial activity in a series of reactions, resulting in the production of hydrogen peroxide (H_2_O_2_) molecules which penetrate the membrane, causing fatal damage. Zinc oxide has led to the reduced utilization of antibiotics in feed, enhancing intestinal epithelial morphology and performance which affects the absorption and digestion of nutrients in broilers [108]. The three major components of intestinal mucosa, epithelial cells, mucus secreting goblet cells (GC), and intraepithelial lymphocytes (IELs), provide a barrier for the entrance of harmful microbes from luminal contents to the underlying capillary network [109]. Intestinal architecture is influenced by zinc supplementation, by increasing the villus height [108]. Zinc also play a key role in the development of the immune system and helps to improve both cellular and humoral immune responses [110] Figure 4. 

Probiotics can replace antibiotics by changing the intestinal micro-biome, thereby producing some of the effects of antibiotics. For example, feed supplementation with probiotics improves feed efficiency and intestinal health and, ultimately, facilitates the faster growth of broilers by reducing intestinal pH, altering the intestinal bacterial composition, and improving digestive activity [111]. Probiotics decrease colonization by intestinal bacteria and, ultimately, stimulate the immune response. Probiotics stimulate endogenous enzyme production, which reduces the production of toxic substances and increases vitamins and/or antimicrobials such as bacteriocins [112]. It has been reported that bacteriocins inhibit the production of toxins and the adhesion of pathogenic microbes [113].

The administration of *Enterococcus faecium*, *Streptomyces* spp., and *Bacillus* spp. in chicken feed triggers antibacterial effects on other pathogenic bacteria in the small intestine. They are essential in improving the water-holding capacity, color, pH, oxidation stability, and the chemical composition of meat, such as the fatty acid content [114]. Feed supplemented with *B. licheniformis* improves the juiciness, flavor, and color of broiler chicken meat, which is appreciated by consumers [115]. Probiotic supplementation also lessens parasitic infestation in chickens [116]. Probiotics exert coccidiostatic effects on *Eimeria tenella*, maintain intestinal health, and reduce the spread and risk of coccidiosis in broiler production systems [115].

## 6. Hepatoprotective Activity of Zinc Nanoparticles

The second-most abundant micro-element contained in the body of animals is zinc (Zn). It cannot be stored in the body and must be consumed on a regular basis to satisfy physiological demands [117]. Zinc is primarily associated with carbonic anhydrase in red blood cells, where it makes up 80–90% of the zinc in whole blood. Other proteins and free amino acids also bind to zinc in plasma, but most of it is attached to albumin. Zinc ion states constitute only a small percentage of plasma zinc. Mineralized zinc is found in the bone, kidney, liver, pancreas, retina, and prostate among other tissues in the body [118]. Zinc deficiency in chicken feed has a negative impact on growth, feed consumption, and the production of insulin-like growth factor-I, growth hormone binding protein, and growth hormone receptor. Zinc is typically found in chicken diets as inorganic zinc pelleting, zinc chloride, zinc oxide, or organic acid and amino acid chelate. Zinc oxide nanoparticles are also used in feed ingredients as a feed additive. By improving growth and feed efficacy, these nanoparticles meet all of the body’s basic demands. It also raised the levels of total protein, glucose, cholesterol, and albumin [119,120,121,122].

The liver is largest gland in the animal body having specialized cells known as hepatocytes that are involved in bile production and serve as a storage site for glycogen, vitamins, and minerals. A liver gland performs several functions such as protein production and drug metabolism, i.e., modification/transformation of a chemical substance into a less toxic and simpler form by enzymatic process [123].

## 7. Anticancer Activity of Zinc Nanoparticles

The number of newly diagnosed cancer patients has increased significantly over the past few years, affecting both their physical and mental well-being. This has led to many patients experiencing high levels of anxiety, distress, and depression [124]. A WHO report estimates that 10 million people lost their life to cancer in 2020, making it a major impediment to increasing life expectancy. In the next two decades, cancer-related diagnoses are expected to exceed 30 million each year, with 16.4 million cancer-related fatalities [125]. There are no discernible changes between the chemotherapeutic regimens of malignant and normal cells, resulting in systemic toxicity and adverse effects. As a result, a novel class of chemotherapeutic drugs with great selectivity for cancer cells is required [126]. The use of anticancer drugs, laser, radiation, and hormonal therapies often causes some abnormalities in the body of the patient. Their toxic effects could harm normal cells and essential organs, resulting in diminished health and lowering life quality [127] Table 2. 

Metal NPs are extremely promising for detecting, diagnosing, and treating cancer [128]. The anticancer activity of zinc oxide nanoparticles is significant despite its high toxicity, solubility, and better bio-availability than solo agents. Cell viability and inhibition increase with an increasing concentration of zinc oxide nanoparticles [129]. ZnO NPs have an electrostatic characteristic that allows them to deliberately target cancerous cells without damaging normal cells [130,131]. The abundant presence of negatively charged phospholipids on the surface of cancer cells causes electrostatic interaction with zinc oxide nanoparticles. This enhances nanoparticle uptake by cancer cells, resulting in cytotoxicity [132]. 

The mechanism by which metallic nanoparticles combat cancer is quite complex and is still being investigated. Cancer is a disease characterized by uncontrolled cell proliferation [133]. A cancer cell normally exists in the body, functioning like a normal cell. It is caused by unregulated cell division, which leads to uncontrolled proliferation of cells. Cancer cells are capable of developing their own blood supply, becoming detached from the original organ, migrating to other organs, and spreading during their development phase [134]. 

The semiconducting characteristics of biologically synthesized zinc oxide nanoparticles have been demonstrated to cause the generation of ROS on the particle’s surface. Through direct interactions between nanoparticles and a cancer cell membrane, this causes oxidative stress and, ultimately, cell death in cancer cells [132]. Nanoparticles of ZnO may induce cellular degradation, loss of membrane integrity, and DNA damage [135]. Cinnamon oil NE encapsulated with vitamin D inhibits the cell cycle progression in the G0/G1 phase and changes the protein expression of Bcl2. This increased the number of apoptotic cells and decreased mitochondrial membrane potential [136]. Reactive oxygen agents are likely to cause cell cycle arrest throughout the growth and preparation for mitosis phases, as well as the meiosis phase [137]. They are thought to work by raising the ratio of B-cell lymphoma protein 2-associated X to B-cell lymphoma protein, which determines cell death [137,138,139]. Zinc benefits p-53 (the tumor suppressor gene) by activating caspase-6 (Casp6), an enzyme that is responsible for orchestrating apoptosis [132,140]. In addition, caspase-3 and 9 play side roles in apoptotic activity, as shown in Figure 5. 

The study by [141] sought to synthesize ZnONPs from *Rosa canina* plant extracts to improve resistance to alveolar adenocarcinoma cells (A549). At 0.1 mg-mL^−1^ of ZnO nanoparticle concentration, 73% of cancer cells were destroyed, which suggests synthesized ZnO nanoparticles contribute significantly to cancer treatment. In the same way, ref. [142] used leaf extracts from *Ziziphus nummularia* to synthesize zinc oxide nanoparticles to treat Hela cancer cases. When the leaf extract concentration was raised from 2 to 200 g/mL of *Ziziphus nummularia* leaf extract, the cell viability of Hela cancer cells reduced from 96% to 60%. Conversely, ZnO nanoparticles showed a dose-dependent cytotoxic effect, i.e., the viability of cells decreased drastically when the concentration of ZnO nanoparticles increased. By synthesizing ZnO nanoparticles, cell viability was reduced from 89% to 39%. This clearly showed that ZnO nanoparticles had a far greater cytotoxic impact than the leaf extract. 

Additionally, ref. [69] have successfully synthesized hexagonal ZnO nanoparticles from leaf extracts of *Mangifera indica.* Obtained nanoparticles had an average size of 50 nm and created cytotoxicity against lung cancer A549 cell lines in a dose-dependent manner. In the same way, other studies conducted by [143,144] synthesized zinc oxide nanoparticles using a precursor salt solution of zinc nitrate hexahydrate. *Anacardium occidentale* leaf extract was able to successfully inhibit the growth of two cancer cell lines (Panc-1 and AsPC-1), working similarly to an aqueous portion of *Gracilaria edulis* against cervical carcinoma cells. Consequently, zinc oxide nanoparticles are more likely to destroy cancer cells in the pancreas, as well as in the cervical lining (siHa cells). ZnO nanoparticles were significantly more toxic in both studies than plant extracts.

**Table 2 molecules-28-01064-t002:** Anticancer activity of biogenic zinc nanoparticle against different cell lines.

Sr.No	Plant Species	Nanoparticles Synthesized	Plant Part	NPs Size (nm)	Shape/ Morphology	Anticancer Activity	Cell Lines Used	Results	Reference
(1)	*Rosa canina*	ZnNO_3_	Fruit	30 nm	Spherical	0.1 mg·mL^−1^	Alveolar adenocarcinoma (A549) cells	Toxicity to A549 cells was dose-dependent	[145]
(2)	*Ziziphus nummularia*	ZnNO_3_	Leaf	12–25 nm	Irregular and spherical	2 and 200 µg·mL^−1^	Hela cancer cell lines	Hela cancer cells lines showed dose-dependent toxicity	[142]
(3)	*Mangifera indica*	ZnNO_3_	Leaf	50 nm	hexagonal	25 µg·mL^−1^	Lung cancer A549 cell lines	A549 lung cancer cells possessed significant cytotoxicity	[69]
(4)	*Costus pictus*	ZnNO_3_	Leaf	40 nm	Hexagonal and rectangular	50 µg·mL^−1^	Daltons lymphoma ascites (DLA) cells	DLA-bearing mice cell lines displayed significant anticancer properties	[67]
(5)	*Anacardium occidentale*	Zn(NO_3_)_2_•6H_2_O	leaf	30 nm	hexagonal	40 µM (Panc-1) and 30 µM (AsPC-1)	human pancreatic cancer cell lines (Panc-1 and AsPC-1)	Toxicity against human pancreatic cancer cell lines was concentration-dependent	[143]
(6)	*Gracilaria edulis*	Zn(NO_3_)_2_•6H_2_O	aqueous	20–50 nm	Hexagonal (Wurtzite) rod	35 µg·mL^−1^	Cervical carcinoma cells (SiHa cells)	Cells of SiHa displayed dose-dependent cytotoxicity	[144]
(7)	*Artocarpus heterophyllus*	Zn(NO_3_)_2_•6H_2_O	Leaf	12–24 nm	Spherical	20 µg·mL^−1^	MDA-MB231 breast cancer cell lines	Dose-dependent nanoparticles suppress breast cancer cell proliferation and induce cytotoxicity	[146]
(8)	*Cucumis melo inodurus*	Zn(CH_3_CO_2_)_2_	Peel	25–40 nm	Crystalline but nearly rounded	40 µg·mL^−1^ (MCF7)	Human (Michigan Cancer Foundation-7 [MCF7])	Associated with the induction of apoptosis in human breast cancer cells (MCF7)	[147]
(9)	*Raphanus sativus*	Zn(CH_3_CO_2_)_2_	leaf	65 nm	spherical	40 µg·mL^−1^	Lung cancer cell line (A549)	Cell viability has been reduced, indicating improved anticancer efficacy.	[148]
(10)	*Pongamia pinnata*	Zn(CH_3_COO)_2_•2H_2_O	seed	30–40 nm	Spherical	50 µg·mL^−1^	Human MCF-7 breast cancer cell lines	Inhibits human MCF-7 breast cancer cells more effectively	[149]
(11)	*Trianthema portulacastrum*	ZnSO_4_	Root, leaf, stem, flower, fruit	20–100 nm	Spherical	100 µg·mL^−1^	Mouse pre-osteoblast cell line (MC3T3-E1)	The cells were found to be viable and showed no toxicity	[150]

## 8. Bio-Imaging Application of Zinc Nanoparticles

The nano-sized zinc nanoparticle surface contains numerous functional groups providing effectiveness in disease diagnosis, drug delivery, bio-imaging, and medication of disease [151]. By targeting cancerous cells, ZnONPs initially loaded with drugs bind to receptors located on the outer surface of the plasma membrane and then release the drugs. Commonly, the site infected by cancers cell has a slightly more acidic PH in nature compared to normal cells. The advancement in drug delivery made by the surface modification of ZnONPs is safe, cheap, stable, eco-friendly, and highly luminescent [152]. Sol-gel approaches are often the best method for altering ZnO quantum dots, since the synthetic reactions are not damaging to ZnO surface defects and wet chemical procedures have been extensively established [153]. 

There are a few bio-imaging techniques, such as computed tomography, magnetic resonance imaging, positron outflow tomography, and fluorescence imaging. These techniques present various benefits and in this way are utilized in different applications. For example, in live cells, fluorescence microscopy provides a valuable tool for monitoring cell physiology. In fluorescent microscopy, the fluorophores are generated by the bio-labelling of a sample or nanoparticles after incubation. Thus, these fluorophores emit light usually to visualize the translational modification in different cell sites and are also involved in the identification of protein complex formation [154,155]. Keeping that in view, in sol-gel techniques, various functional groups or sometimes entire chemicals, such as polyvinylpyrolidone (PVP), oleic acid (OA) substituted with diethanolamine (DEA), and polyethylene glycol methyl ether (PEGME), were used as capping agents on ZnONPs, but such chemicals had no significant biomedical applications [95,156,157]. In contrast, fluorophores and nano-probes are effective in cancer therapy and in the treatment of cardiovascular disease in animals, without damaging other normal cells nearby [158,159] Figure 6.

The uptake of nanoparticles by cells is determinant on the following factors: size; charge; nanoparticle conjugation with protein; and ligand, oligodeoxynucleotide, and endosome aggression [160,161,162]. The same chemical composition in smaller nanoparticles is more effective at bypassing degradation pathways than in larger ones [163]. In the modern era, magnetic nanoparticles have recently received a lot of attention due to their advancement in picture directed treatment (e.g., fluorescence, MRI, X-ray CT). During treatment, the MRI generating heat and reactive oxygen species (ROS) target infected tissues or cells to eliminate cancerous cells from the endosome and other organelles [164]. 

## 9. Drug Delivery

Zn was initially employed as a biomedical ingredient in medicinal skin lotions in ancient Egyptian times. During Roman times, zinc was widely utilized in cosmetics and lotions to treat skin ailments and mend wounds. ZnO was the main ingredient in “luna fixa,” a secret remedy devised by alchemist Ludemann to alleviate convulsions and spasms [129]. A chemical (zinc sulphate) was a medicinal agent prescribed by Paracelsus, the pioneer of pharmacology, which led to powerful advancements in biomedical research. Several biological activities rely on zinc, such as fetal development, natural growth, wound healing, metabolism, immunity, cognitive functioning, sperm production, bone mineralization, neurological function, and enzymatic processes [165]. ZnO nanoparticles are excellent nanoplatforms for bio-imaging and drug delivery, among other uses, due to their variable surface chemistry, wide surface area, and photo-toxic impact. In vitro studies have shown that ZnO nanoparticles are very hazardous to cancer cells, microbes, and leukemic T cells [75]. 

In addition to its regulatory role in the processes of normal growth and reproduction, zinc (Zn) is also vital for wound healing, ossification, DNA synthesis, cell division, and gene expression. Zinc supplementation improves animal immune systems through lymphocyte multiplication and antibody production [166]. In addition to nanospheres, nanosheets, nanorods, nanobelts, and quantum dots, ZnONPs are fabricated in various shapes and sizes for drug delivery systems [167]. NP-mediated targeted drug delivery involves ligand-receptor recognition, hydrophobic and coulombic interactions, etc. Abiotic factors such as pH, light, temperature, and light intensity influence the release of drugs and their response. Since pH values in tumors and immune-mediated tissues are significantly lower than those in blood and normal tissues, pH-responsive drug delivery systems are easy to use and have outstanding design advantages [168]. 

Furthermore, ZnO QDs exhibit traits such as the capacity to generate ROS, good adsorption capabilities, and a highly tailored surface, making them an attractive option for usage in biomedical applications. ZnO crystals are suitable for photo-catalytic treatment because they produce ROS when exposed to UV light in an aqueous medium. The surface valence band holes of a ZnO QD can absorb electrons from water and/or hydroxyl ions, resulting in the generation of hydroxyl radicals (OH•). In addition, the reduction of oxygen produces the superoxide anion O^2−^ [152]. In addition to the substantial creation of ROS by UV exposure, ZnO QDs can also produce minor amounts of ROS when they interact with nanoparticles. The use of a drug delivery system (DDS) in nano-medicine, providing a platform for targeting cancerous cells, has replaced the conventional method on the basis of certain features that include lower toxicity, higher bio-availability, and increased solubility in cells. 

The authors of [169] demonstrated that ZnO quantum dots (QDs) could be used to inhibit the pore-opening of doxorubicin (DOX)-loaded mesoporous silica nanoparticles (MSN). DOX was successfully loaded into channel-like pores within MSN, followed by the use of NH2-ZnOQDs to seal the pores via covalent amide bonds formed with carboxylic acid groups attached to the outer surface of MSN. ZnOQD lids were rapidly dissolved in the weakly acidic intracellular compartment after the internalization of the nanoassemblies in HeLa cells, resulting in DOX being released into the cytosol. Aside from that, ZnONPs also produced synergistic anti-tumor effects in HeLa cells due to their intrinsic anticancer properties. 

Our most recent study [139] proposed a novel type of a ZnO polymer–DOX system that is biodegradable and has a high loading capacity of more than 20% wt. Because the ZnO polymer–DOX combination could be dissolved under acidic conditions, more than 90% of the DOX deposited onto the ZnO surface was liberated in buffer solutions with a pH of 5.0. ZnO QDs were covered with biodegradable polymer shells that were non-toxic to human brain cancer cells U251. The toxicity of DOX-loaded ZnO nanoparticles should theoretically be lower than that of DOX alone; however, at acceptable concentrations, the ZnO–DOX composites displayed even greater cytotoxicity. The augmentation of drug adsorption was attributed to cytotoxicity enhancing actions by the nanocarrier. Unbound DOX molecules entered cells by passive diffusion, and there was a saturation concentration of DOX inside some drug-resistant cells, such as U251. When the DOX concentration outside the cells was gradually raised, the DOX cytotoxicity toward U251 reached a peak (about 40 percent of cell viability). However, when ZnO nanocarriers were used, the situation altered. When ZnO–DOX composites were taken up by cells, they were absorbed by endosomes and lysosomes, preventing DOX saturation in the cellular fluid; hence, more and more ZnO–DOX was constantly taken up. Finally, the ZnO–DOX composites degraded in the lysosomes, releasing significant amounts of DOX molecules and demonstrating greater cytotoxicity. DOX fluorescence was observed throughout the cytoplasm after 3 h of incubation. Furthermore, further CLSM observations demonstrated that DOX molecules penetrated the nucleus to damage DNA, and that Zn^2+^ ions were released from lysosomes and enriched by zincosomes in the cytoplasm (Zhang et al., 2013). Indeed, Nel and colleagues thoroughly investigated the dissolving behavior of ZnO nanoparticles at consecutive nano-biological boundaries outside cells as well as the acidic environment of lysosomes. They discovered that releasing Zn^2+^ ions into cells might have a number of negative consequences, including lysosomal degradation, mitochondrial disruption, and ROS production. As a consequence, ZnO nanoparticles devoid of particular manipulation showed substantial cytotoxicity. However, if ZnO nanoparticles are securely covered with insulating shells, they will be highly durable in pH 7.4 buffer solutions and cell cultures, and will not disintegrate outside cells. The medications or biomolecules can be adsorbed on the ZnO surface or embedded in the shells around ZnO QDs, and they will be securely delivered into cells by ZnO nanoparticles, where drugs will be released to execute their functions [170,171] fabricated porous ZnO nanorods functionalized with folic acid for targeted, pH-sensitive DOX delivery. Because the nanocarrier-loaded with DOX was more effective in the uptake of folates by MDA-MB-231 cells over-expressed with the folate receptor, it showed higher cytotoxicity. Furthermore, no systemic toxicity of the nanocarrier was identified in vivo Figure 7. 

## 10. Conclusions

Overall, from the studies reviewed, it is considered that plant-mediated ZnONPs synthesis is significantly safer and more ecologically friendly than physical and chemical methods. Because of their relevance and adaptability, ZnONPs have a wide variety of applications in the industrial, agricultural, and poultry industries. On the other hand, their biomedical applications are increased day by day in various processes including bio-imaging, drug delivery, biosensors, and gene delivery. With respect to its toxicity properties, ZnONPs can act as smart weapons against multiple drug-resistant microorganisms and as a talented substitute for antibiotics. It is anticipated that this review could further streamline the research on innovative methodological and clinical correlations in this area.

## Figures and Tables

**Figure 1 molecules-28-01064-f001:**
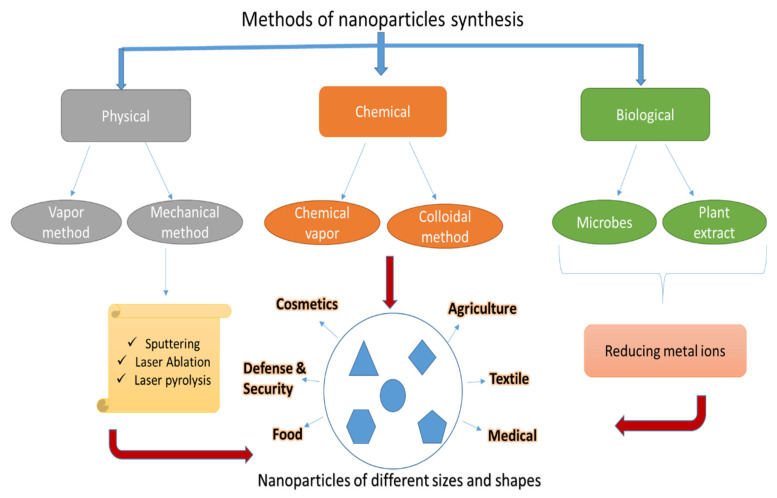
Different techniques for synthesis of nanoparticles including: (physical), which is size and shape dependent; (chemical), the synthesis of NPs from precursor chemical such as polyvinyl alcohol, ethylenediamine tetraacetic acid (EDTA), oliec acids and many others; and (biological) synthesis from plants, bacteria, and fungi. Properties associated with physical, chemical, and biological synthesis of NPs.

**Figure 2 molecules-28-01064-f002:**
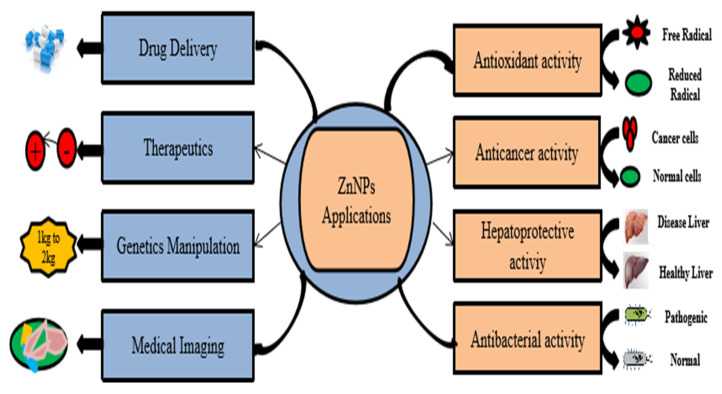
Application of Zinc nanoparticles on health performance of broilers. Antioxidant activity (Zn NPs responsible for conversion of free radical to reduced radical), anticancer activity (cease or slow production of diseased cells), hepatoprotective activity (diseased liver to healthy liver), antibacterial activity (against bacterial infections), targeted drug delivery in dose-dependent manner, genetic manipulation (boosting body weight), and medical imaging (to obtain a pectoral view of targeted cell or animal tissues).

**Figure 3 molecules-28-01064-f003:**
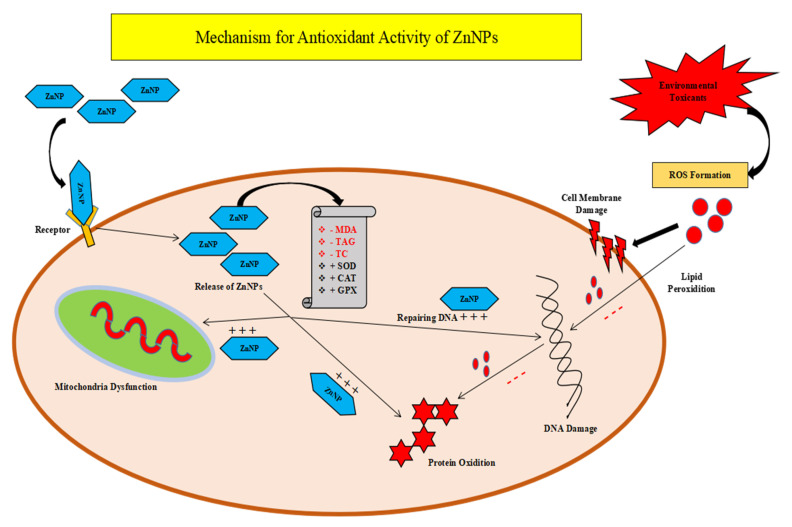
Initially Zn NPs attached to membrane bounded receptors and are able to enter animal cells. Cell membrane damage, lipid peroxidation, protein oxidation, DNA damage, and mitochondrial dysfunction caused due to environmental factors medicated through Zn NPs. Enzyme level increase in animal cell, i.e., superoxide dismutase (SOD), catalase (CAT), glutathione peroxidase (GPX), but a reduction in enzyme level of malondialdehyde (MDA), total antioxidant capacity (TAC), and total cholesterol.

**Figure 4 molecules-28-01064-f004:**
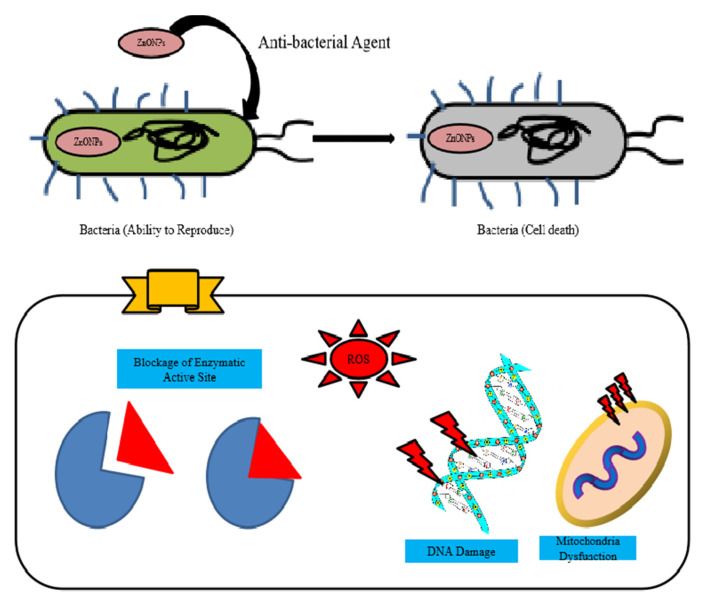
Antibacterial activity of ZnO nanoparticles. Upon attachment and entry to bacterial cell there are three possible ways through which bacterial cells are killed. ZnO nanoparticles either block the electron transport chain which results in low production of ATP, or they interact with various enzymes, resulting in enzyme alteration due to which the substrate cannot bind to enzyme, hence resulting in low concentration of product. Thirdly, they produce reactive oxygen species (ROS) which cause oxidative damage to different biomolecules such as DNA, proteins, and lipids, resulting in altered cellular machinery.

**Figure 5 molecules-28-01064-f005:**
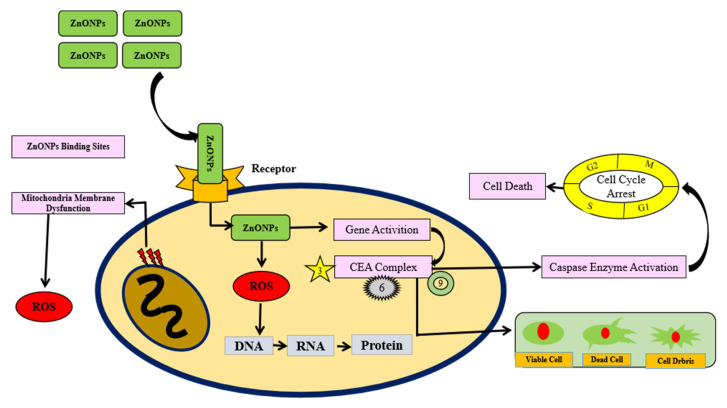
ZnONPs arrest the cancer cell cycle by generating ROS in effected cells. ZnONPs binds to receptors on the plasma membrane to enter the cell. ZnONP creates reactive oxygen species (ROS) in cytoplasm and then interaction with DNA brought structural changes in proteins. Secondarily, ZnONPs activate (p-53) genes via caspase enzyme activation (CEA) that arrest cancerous cell cycle at synthesis phase (S-phase) leading to cancer cell death.

**Figure 6 molecules-28-01064-f006:**
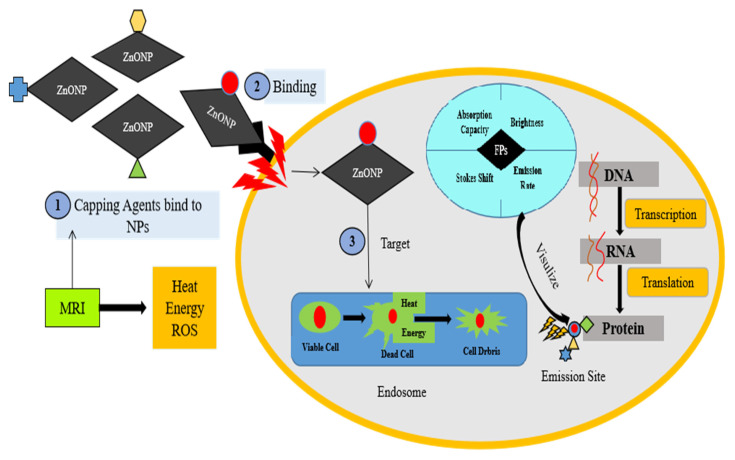
Demonstrated that entrance of ZnO NPs in animal cells required heat energy. Fluorescent probes associated with ZnO NPs act as capping agents damaging the plasma membrane of cells. Endosomes are responsible for excretion of cell debris. Fluorescent probes appeared on target tissue or attached on the surface of proteins.

**Figure 7 molecules-28-01064-f007:**
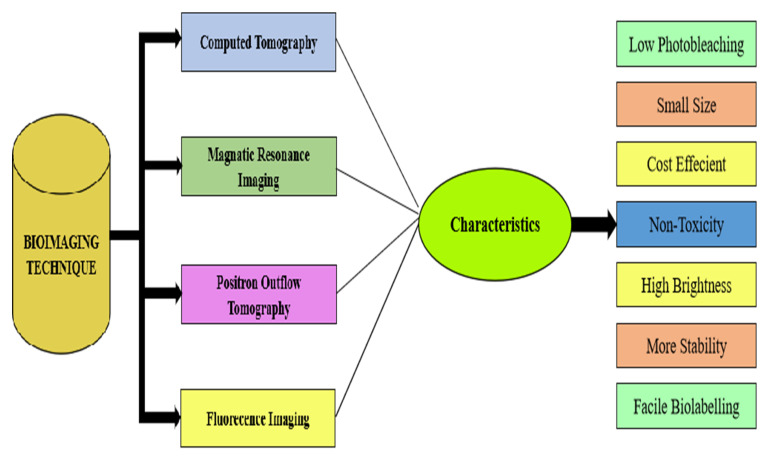
Left side shows different techniques used for bio-imaging. Right side demonstrates various feature associated with bio-imaging.

**Table 1 molecules-28-01064-t001:** Different Antioxidant enzymatic activities in various cells and organism on exposure to ZnO NPs.

Size	Assay/Cell Type/Animal Model	Antioxidant Effects	Reference
40–50 nm	DPPH	Time and concentration dependent radical scavenging activity was reported	[66]
10–20 nm	DPPH assay	ZNONPs at concentration range of 1.5–12.5 mb/mL showed radical scavenging activity	[67]
33–73 nm	DPPH assay	DPPH free radical scavenging activity was noted at concentration range of 0.125–1 mg/mL	[68]
40–60 nm	DPPH assay	ZnO NPs at concentration of 10–50 μg/mL showed antioxidant activity	[69]
5 nm	DPPH Assay	Biologically prepared ZnO NPS showed radical scavenging and cytoprotective activities under induced oxidative stress in adipocytes	[70]
39.2 nm	Broilers	ZnO NPs injected at 40 and 80 ppm resulted in increased antioxidant activity with reduced MDA content	[71]
40 nm	SOD, POD, CAT and GPX	Dose-dependent antioxidative properties were found in both brain and liver tissues	[72]
20 nm	Egg laying hens	Enhanced superoxide dismutase activity in liver Decreased MDA content in eggs	[73]
20 nm	Egg laying hens	Enhanced SOD activity in pancreas and plasma Reduction in MDA content was observed in eggs	[73]
35–45 nm	Broilers	Reduction in MDA content in plasma and liver was observed when birds exposed to low ambient temperature as well	[73]

## Data Availability

Not applicable.

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
