# Peer review of "Mechanistic Approaches to the Application of Nano-Zinc in the Poultry and Biomedical Industries: A Comprehensive Review of Future Perspectives and Challenges"

_molecules, 2023, doi:10.3390/molecules28031064_

Round 1

Reviewer 1 Report

This review work “Zinc Nanoparticles as Feed Additives for Poultry. A Compre-hensive review on Future Perspectives and Challenges” is interesting and sounds well. This review discussed the formulation and recent advancements of zinc oxide nanoparticles from plant sources and their anti-can-cerous activities, wound healing, and drug delivery, followed by a detailed discussion of their mechanisms of action. Although this topic is good, this article lacks the author's own point of view, and some format errors and writing style are not specification.

Specific points:

1In figure 1. You should give different processing key parameters and typical processes to help readers better understand the preparation of nanoparticles.

2)“Moreover, many factors such as temperature, pH, metal precursor con-centration, and reaction time are significant in determining the rate of formation, yield, and morphology of NPs.Could you please give some examples?

3Regarding to the Antioxidant Activity of Zn NPs, could you please summarize this research in a Table?

4)“Numerous studies relate oxidative stress to heat stress, implying a simultaneous increase in cell death and ROS formation in certain cells So, please give more examples

5Figure 4 is too simple and not strong scientific, you should improve this or delete

6Figure 6. Could you please give some pictures about Bio-imaging?

7”Our most recent study (Zhang et al., 2013)” so 2013 is “most recent”?

8Outlooks must be added before the “Conclusion” section

Author Response

Dear Reviewer 1,

Thank you for reviewing our manuscript entitled ‘’ Mechanistic approaches to the application of nano-zinc in poultry and biomedical industry: A Comprehensive review on Future Perspectives and Challenges’’ we tried our best to incorporate the comment/suggestion as you recommend in the revision of manuscript.

Comment-1: In figure 1. You should give different processing key parameters and typical processes to help readers better understand the preparation of nanoparticles.

Response: We are thankful for raising this response, as the previous figure was very simple now we have replaced it with new figure which thoroughly elaborates about the various synthesis methods of nanoparticles along with the resulting different sizes and shapes of nanoparticles and their applications in different sectors. highlighted in track mode changes.

Comment 2: “Moreover, many factors such as temperature, pH, metal precursor con-centration, and reaction time are significant in determining the rate of formation, yield, and morphology of NPs.”Could you please give some examples?

Response: As nanoparticles synthesis is a reduction process in which salts of different metals are reduced to their respective nanoparticles, this reduction method is affected by various factors mainly by pH and temperature. For example, temperature and pH have a major effect on shape and size of the nanoparticles. These factors and their effects on nanoparticles synthesis have now been discussed in the revised manuscript file. highlighted in track mode changes.

Comment-3: Regarding to the Antioxidant Activity of Zn NPs, could you please summarize this research in a Table?

Response: As we know that antioxidant activity is one of the major characteristic of ZnO NPs, in the manuscript we have discussed it thoroughly in description but its literature review was missing, which we have now added in the Table form in the revised MA script file .as highlighted in track mode changes.

Comment-4: Numerous studies relate oxidative stress to heat stress, implying a simultaneous increase in cell death and ROS formation in certain cells” So, please give more examples

Thanks for raising this comment, as cells are prone to various types of abiotic stresses which leads to ROS accumulation and resulting cell damage. Heat stress is also responsible for production of ROS which ultimately leads to cell damage and cell death. Discussion about heat stress related cell death is now discussed in revised manuscript file at appropriate place as highlighted in track mode changes.

Comment-5: Figure 4 is too simple and not strong scientific, you should improve this or delete

Response: As antibacterial activity is attributed to ZnO nanoparticles, mechanistic diagram is now added in the revised manuscript. highlighted in track mode changes.

Comment-7: ”Our most recent study (Zhang et al., 2013)” so 2013 is “most recent”?

Response: We great full to reviews for raising this comment, this citation is now replaced with latest and appropriate study as highlighted in track mode changes.

Reviewer 2 Report

In your manuscript, you described different techniques for the synthesis of nanoparticles. It is recommended that you state in a few sentences which technique is the best and why you would recommend it over the others.

The subtitle "Application of ZnO NPs" is too broad. Since the title of the paper is "Zinc Nanoparticles as Feed Additives for Poultry. A Comprehensive review on Future Perspectives and Challenges" should describe in more detail the application of ZnO NPs in poultry production.

It is not necessary to describe the application of ZnO NPs in other industries.

In general, a large part of the paper is not closely related to the use of nanoparticles in poultry feed. According to the title of the paper, it is expected that there will be a part that talks about the influence of the addition of these particles on the productive properties of poultry, reproduction, immunity and the like. Maybe consider changing the title of the manuscript.

Author Response

Dear Reviewer 2,

Thank you for reviewing our manuscript entitled ‘’ Mechanistic approaches to the application of nano-zinc in poultry and biomedical industry: A Comprehensive review on Future Perspectives and Challenges’’ we tried our best to incorporate the comment/suggestion as you recommend in the revision of manuscript.

Comment: In your manuscript, you described different techniques for the synthesis of nanoparticles. It is recommended that you state in a few sentences which technique is the best and why you would recommend it over the others.

Response: we are thankful to th reviewer the for raising such in an important issue, extraction information has been added to the MS according to the need and to maintain the beauty of the MS. The changes has been highlighted in track mode changes.

Comment: The subtitle "Application of ZnO NPs" is too broad. Since the title of the paper is "Zinc Nanoparticles as Feed Additives for Poultry. A Comprehensive review on Future Perspectives and Challenges" should describe in more detail the application of ZnO NPs in poultry production.

Response: we tried a lot to find out published papers related to the application of Zinc nanoparticle in poultry feed, we tried to use and cite all published material but unfortunately no more data has been published till now. So we than go for general uses. But now the topic of the MS has been modified according to the need of the study. The changes are highlighted in track mode changes.

Comment: it is not necessary to describe the application of ZnO NPs in other industries.

Response: we have change our topic so that we could cover more used of zinc in poultry also in other industry which are linked to the poultry industry. The changes are highlighted in track mode changes.

Comment: In general, a large part of the paper is not closely related to the use of nanoparticles in poultry feed. According to the title of the paper, it is expected that there will be a part that talks about the influence of the addition of these particles on the productive properties of poultry, reproduction, immunity and the like. Maybe consider changing the title of the manuscript

Response: we are thankful to the reviewer for highlighting this issue, the topic of manuscript is now changes and modified according to manuscript description and need as highlighted in track mode changes.

Round 2

Reviewer 1 Report

Accepted